# Ursolic Acid Enhances the Sensitivity of MCF-7 and MDA-MB-231 Cells to Epirubicin by Modulating the Autophagy Pathway

**DOI:** 10.3390/molecules27113399

**Published:** 2022-05-25

**Authors:** Zhennan Wang, Pingping Zhang, Huan Jiang, Bing Sun, Huaizhi Luo, Aiqun Jia

**Affiliations:** 1School of Environmental and Biological Engineering, Nanjing University of Science and Technology, Nanjing 210094, China; 15050525636@163.com (Z.W.); huanjiang929@foxmail.com (H.J.); yangwenleg@163.com (B.S.); rohizer@163.com (H.L.); 2State Key Laboratory of Marine Resource Utilization in South China Sea, School of Pharmaceutical Sciences, Hainan University, Haikou 570228, China; bbsuju2@163.com

**Keywords:** breast cancer, autophagy, drug sensitivity, ursolic acid, epirubicin

## Abstract

Breast cancer is the leading cause of cancer death among women in the world, and its morbidity and mortality are increasing year by year. Epirubicin (EPI) is a commonly used drug for the treatment of breast cancer but unfortunately can cause cardiac toxicity in patients because of dose accumulation. Therefore, there is an urgent need for new therapies to enhance the sensitivity of breast cancer cells to EPI. In this study, we found ursolic acid (UA) can significantly improve the drug sensitivity of human breast cancer MCF-7/MDA-MB-231 cells to EPI. Next, we observed that the co-treatment of UA and EPI can up-regulate the expression of autophagy-related proteins Beclin-1, LC3-II/LC3-I, Atg5, and Atg7, and decrease the expression levels of PI3K and AKT, which indicates that the potential mechanism should be carried out by the regulating class III PI3K(VPS34)/Beclin-1 pathway and PI3K/AKT/mTOR pathway. Furthermore, we found the autophagy inhibitor 3-methyladenine (3-MA) could significantly reverse the inhibitory effect of co-treatment of UA and EPI on MCF-7 and MDA-MB-231 cells. These findings indicate that UA can dramatically enhance the sensitivity of MCF-7 and MDA-MB-231 cells to EPI by modulating the autophagy pathway. Our study may provide a new therapeutic strategy for combination therapy.

## 1. Introduction

Breast cancer is a devastating disease mainly caused by malignant lesions of the ductal epithelium of the breast, and it affects 2.1 million women worldwide each year, accounting for about 14% of cancer-associated deaths [1]. Epirubicin (EPI) (Figure 1B) is an anthracycline drug commonly used in the clinical treatment of breast cancer. Unfortunately, it exhibits a serious side effect—cardiotoxicity, which is caused by dose accumulation [2], and it also damages other human organs such as the brain, kidney, and liver [3]. The resistance of cancer cells to current anticancer drugs and the side effects of drugs are still obstacles to successful cancer treatment [4]. Therefore, it is necessary to develop new anticancer agents or new therapeutic strategies.

Autophagy plays a crucial role in cell survival and cancer development, which is a self-regulatory mechanism widely existing in eukaryotic cells [5]. The process of autophagy is related to a variety of signaling pathways, such as class I and class III phosphoinositide 3-kinase (PI3K) pathways: the class I PI3K/AKT/mTOR pathway could control the phosphorylation and suppression of ULK1, a mammalian serine/threonine protein kinase that plays a key role in the initial stages of autophagy [6], while the class III PI3K(VPS34)/Beclin-1 pathway is essential for initiating autophagy [7]. Interestingly, the regulation of autophagy involves diverse signaling pathways, which are also involved in tumorigenesis [8]. Recent studies indicate the therapeutic value of autophagy as a novel target for anticancer therapy [7,9].

Nowadays, more and more agents from natural products have been used to treat cancer based on combination therapy [10], such as red guava extracts, which could promote doxorubicin-, tarceva-, and iressa-induced cytotoxicity in triple-negative breast cancer through modulating apoptotic or necrotic cell death pathways [11]. Ursolic acid (UA) (Figure 1A), a pentacyclic triterpene, is a common chemical constituent existing mainly in the Lamiaceae family [12]. Studies have indicated that UA can affect multiple molecular signaling pathways related to human carcinogenesis, apoptosis, inflammation, metastasis, autophagy, and proliferation [13,14]. Lewinska et al. found UA could mediate changes in the glycolytic pathway to promote cytotoxic autophagy and apoptosis in phenotypically different breast cancer cells [15]. Therefore, we designed this combination treatment study for EPI with UA to examine whether it potentiates the treatment efficiency on MCF-7 and MDA-MB-231 cells in vitro, and if so, the underlying mechanisms of the combined chemotherapy of UA and EPI would be explored.

In this study, we found that the UA can significantly improve the drug sensitivity of human breast cancer MCF-7 cells and MDA-MB-231 cells to EPI. Furthermore, we explored the potential mechanism of the enhanced effect of UA on the sensitivity of MCF-7 and MDA-MB-231 cells to EPI. The results revealed that co-treatment of UA and EPI can regulate the class III PI3K(VPS34)/Beclin-1 pathway and PI3K-Akt-mTOR signaling pathway, up-regulate the expression of autophagy-related proteins Beclin-1, LC3-II;/LC3, Atg5 and Atg7, and then enhance autophagy. These findings provide evidence that UA might be an effective adjunct to EPI in the treatment of breast cancer.

## 2. Results

### 2.1. Effects of UA and EPI on the Viability of MCF-7/MDA-MB-231 Cells

Firstly, the functions of UA were evaluated on human breast cancer MCF-7 and MDA-MB-231 cells by CCK-8 assay, respectively. As shown in Figure 1C, at concentrations of ≤20 μΜ, UA displayed no ability to suppress the viability of MDA-MB-231 cells within 24 h. Regarding MCF-7 cells, UA exhibited little cytostatic effect at concentrations of 10 μΜ (Table 1), and when the concentrations of UA were 40–80 μM, the viability of MCF-7 and MDA-MB-231 cells were dose-dependently suppressed.

Secondly, we measured the viability of MCF-7 and MDA-MB-231 cells after 24 h incubation with EPI, at concentrations of 0–20 μM. As shown in Figure 1D, EPI could markedly (*p*-value < 0.001) suppress the viability of MCF-7 and MDA-MB-231 cells at the concentrations of 5–20 μM. Compared with the group of the concentration of 10 μM, EPI at the concentrations of 20 μM had little enhancement of the inhibitory activity on the two kinds of breast cancer cells.

### 2.2. Effect of UA on the Sensitivity of MCF-7/MDA-MB-231 Cells to EPI

In order to investigate the effect of UA on the sensitivity of MDA-MB-231 cells to EPI, we chose 10 and 20 μM doses of UA to test the combinative therapy effect with EPI, because of the little inhibitory effect (the viability of cells exceeding 90%) of UA on the MDA-MB-231 cells at those concentrations. For the same reason, we chose 5 and 10 μM doses of UA to conduct the same assay on MCF-7 cells. The results showed UA plus EPI in combination could markedly suppress the proliferation of MCF-7 and MDA-MB-231 cells compared with the control group (Table 1 and Table 2 and Figure 2). When treated with the 5 and 10 μM doses of UA alone, the viability of MCF-7 cells was 98.91% and 93.78%, respectively, which indicated that UA had no significant effects on the viability of MCF-7 cells. Also, the viability of MCF-7 cells was decreased to 92.83%, 83.40%, and 68.67%, respectively, when treated with EPI alone at concentrations of 0.5 μM, 1 μM, and 5 μM. Furthermore, compared with the UA-treated alone or EPI-treated alone groups, the inhibition of the viability of MCF-7 cells was significantly enhanced after treatment with UA plus EPI in combination, and UA exhibited a significantly enhanced effect on the sensitivity of MCF-7 cells to EPI when treated with 10 μM UA plus 1 μM EPI in combination (Table 1). Similar results were also observed in MDA-MB-231 cells. UA showed little inhibitory effect on the viability of MDA-MB-231 cells at the concentrations of 10 μM and 20 μM (the viability of MDA-MB-231 cells was 97.50% and 90.12%, respectively), and UA also showed a remarkable enhanced ability on the sensitivity of MDA-MB-231 cells to EPI when treated with 20 μM UA plus 1 μM EPI in combination (Table 2). Additionally, compared with the control group, there was little difference in activity levels of Caspase 3 in the UA-treated group, EPI-treated group, and UA plus EPI treated group (Appendix A), which indicated the inhibitory effects of UA plus EPI in combination on MCF-7/MDA-MB-231 cells related little to apoptotic pathways.

### 2.3. Effect of UA and EPI on the Migration of MCF-7/MDA-MB-231 Cells

The effect of EPI and/or UA on the migration capabilities of MCF-7 and MDA-MB-231 cells was detected by transwell assay (Figure 3). Compared with control groups, the amount of stained MCF-7 and MDA-MB-231 cells was significantly reduced after treatment with UA plus EPI in combination. Furthermore, compared with the EPI-treated alone group, the number of MCF-7 cells is further reduced remarkably when treated with UA plus EPI in combination, which indicates that UA could enhance the inhibitory effect of EPI on the migration of MCF-7. However, UA did not exhibit a remarkable enhanced effect to EPI on the inhibition of the migration of MDA-MB-231 cells.

### 2.4. Autophagy Is Involved in the Treatment of UA plus EPI in Combination in MCF-7/MDA-MB-231 Cells

It has been reported that autophagy could suppress the proliferation of cancer cells, even leading cancer cells to death [16]. We investigated whether autophagy is involved in the treatment of UA plus EPI in combination in MCF-7 or MDA-MB-231 cells. As shown in Figure 4A, the viability of MCF-7 cells was decreased significantly when treated with 10 μM UA plus 1 μM EPI in combination compared with the control group in the absence of any treatments. Also, 3-MA (1mM), the autophagy inhibitor, showed no remarkable effect on the viability of MCF-7 cells, whereas we found the viabilities MCF-7 cells were increased significantly after the treatment of 3-MA+ UA plus EPI in combination compared with the UA plus EPI in combination group. This indicates that 3-MA (1mM) could significantly reverse the inhibitory effect of treatment with 10 μM UA plus 1 μM EPI in combination in MCF-7 cells, as the viability of cells was restored to 93.43%. Additionally, RAPA slightly influenced the inhibitory effect of treatment with 10 μM UA plus 1 μM EPI in combination in MCF-7 cells. A similar result could be found in MDA-MB-231 cells; 3-MA (1mM) restored MDA-MB-231 cell viability to 88.01%, attenuating the inhibitory effect of treatment with 20 μM UA plus 1 μM EPI in combination (Figure 4B). RAPA also slightly enhanced the inhibitory effect of UA and EPI combination treatment. These results are consistent with those of previous studies. Pan et al. found the inhibitory effect of bufalin on colorectal cancer cells was reversed by treatment with 3-MA [17]. Lee et al. reported 3-MA reversed the cytotoxic autophagic cell death induced by the treatment with combination of pristimerin and paclitaxel in MDA-MB-231 cells [18]. Based on these findings, we inferred that the inhibitory effects of UA plus EPI in combination in MCF-7 and MDA-MB-231 cells were related to the autophagy pathway. 

Furthermore, the autophagic vacuoles were analyzed by MDC labeling in MCF-7 and MDA-MB-231 cells. As shown in Figure 5, compared with the control groups, the treatment of UA plus EPI in combination could markedly increase the amount of MDC-labeled vesicles both in MCF-7 and MDA-MB-231 cells, suggesting the induced effect of the treatment of UA plus EPI in combination on autophagic vacuole formation. Compared with UA plus EPI in combination group, the amount of MDC-labeled vesicles was reduced significantly in the UA+EPI+3MA group both in MCF-7 and MDA-MB-231 cells. This result revealed the induction of the treatment of UA plus EPI in combination on autophagic vacuole formation was inhibited by 3-MA. 

Taken together, the treatment of UA plus EPI in combination in MCF-7/MDA-MB-231 cells was related to the autophagy pathway.

### 2.5. UA Promotes EPI-Sensitivity of MCF-7/MDA-MB-231 Cells by Modulating the Autophagy Pathway

To further prove the effects of UA and/or EPI on the autophagy pathway, we evaluated the expression of the autophagy-related proteins by immunoblotting in MCF-7/MDA-MB-231 cells. As shown in Figure 6, compared with the control, the protein expression levels of Beclin-1, LC3-II/LC3-I, Atg5, Atg7 increased significantly when treated with UA plus EPI in combination in MCF-7 cells, accompanied by a decrease in AKT expression and little change in PI3K expression. Furthermore, the treatment of UA plus EPI in combination could markedly increase the protein expression levels of Beclin-1, LC3-II/LC3-I, Atg5, Atg7 and decrease the expression levels of PI3K and AKT, compared with the EPI-treated alone group in MCF-7 cells. Then, we added 3-MA to MCF-7 cells, alone or with UA and/or EPI treatment. The Western blotting results showed that 3-MA could reverse the effect of the treatment of UA plus EPI in combination on the protein expression levels of Beclin-1, LC3-II/LC3-I, Atg5, Atg7 and AKT. 

Likewise, the treatment of UA plus EPI in combination also could markedly increase the protein expression levels of Beclin-1, LC3-II/LC3-I, Atg5, Atg7 and decrease the expression levels of PI3K and AKT, compared with the UA-treated alone or the EPI- treated alone group in MDA-MB-231 cells (Figure 7). The effect of the treatment of UA plus EPI in combination on the protein expression levels of Beclin-1, LC3-II/LC3-I, Atg5, Atg7, PI3K and AKT also could be reversed by 3-MA. Taken together, these results indicated that UA indeed influenced EPI-sensitivity through the autophagy pathway in MCF-7/MDA-MB-231 cells.

## 3. Discussion

EPI is an anthracycline that can be used to treat breast cancer and exhibits the anti-cancer effect by intercalating into DNA strands to block DNA and RNA synthesis and inhibition of topoisomerase II [19]. However, the application of EPI in the treatment of cancer has been limited due to its serious adverse side effects, particularly dose-related cardiotoxicity, which can cause irreversible congestive cardiac failure in patients [20]. Thus, it is of great significance to explore therapy to enhance the sensitivity of breast cancer cells to epirubicin.

Nowadays, more and more attention had been attracted to the application of natural products to chemotherapy for cancer, which could improve chemotherapeutic efficacy [21], and many combination therapies have been explored successfully for antitumor therapy [22]. Virginie Aires et al. revealed that resveratrol and its metabolites could synergize with SN38 or oxaliplatin on colon cancer cells [23]. UA is a pentacyclic triterpenoid widely found in different medicinal herbs, possessing many biological effects including neuroprotection and antidepressant-like and antitumor activities [24,25]. Xavier et al. found UA could modulate autophagy through the JNK pathway in apoptosis-resistant colorectal cancer cells and induce cell death [26]. In this study, we found that ursolic acid can significantly improve the drug sensitivity of human breast cancer MCF-7 and MDA-MB-231 cells to EPI. So, the result indicated that ursolic acid might be helpful to reduce the clinical dose of EPI in breast cancer patients, thereby decreasing the side effects of EPI. Additionally, we cannot disregard the fact that UA plus EPI in combination could decrease the amount of the migrated cells in MCF-7 cells compared with EPI-treated cells but did not affect those significantly in MDA-MB-231 cells. This difference indicated that the highly invasive phenotype of MDA-MB-231 cells cannot be affected by UA and UA might enhance the drug sensitivity of MDA-MB-231 cells to EPI through other mechanisms.

Currently, autophagy is increasingly recognized as playing an important role in tumor therapy [27], and the anticancer mechanism of autophagy is multifaceted. Autophagy could not only inhibit the growth of precancerous cells by repairing damaged organelles, particularly mitochondrial networks [28], but could also maintain genome stability by clearing off abnormal organelles or protein aggregation to prevent malignant transformation during tumorigenesis [29]. LC3-II protein is a general marker for the formation of autophagosomes due to its role as an important component of the autophagosome membrane [30]. The LC3-II protein assembly process was proposed as shown in Appendix A [28]. The microtubule-associated protein LC3 was proteolytically cleaved to generate the LC3-I precursor molecule under the promotion of ATG4, and the ligases ATG7 and ATG10 facilitated ATG12–ATG5 complex formation and further formed the ATG12–ATG5–ATG16L1 complex that promoted conjugation of LC3-I to PE, producing LC3-II. Thus, the LC3-II/LC3-I ratio was correlated with autophagic flux. Besides, Beclin-1, a crucial mammalian autophagy protein, is involved in the interaction between autophagy and cell death pathway as part of a core complex that contains vacuolar sorting protein 34 (VPS34), a class III phosphatidylinositol-3 kinase [31]. Leng et al. found UA promotes cancer cell death by inducing Atg5-dependent autophagy [32]. Additionally, UA has also been demonstrated to cause the upregulation of Beclin-1 in hypertrophic scar fibroblasts [33]. Similar results were also found in our study. We found the protein expression levels of Beclin-1, LC3-II/LC3-, Atg5 increased significantly after the treatment of UA plus EPI in combination, compared with the UA-treated alone or the EPI-treated alone group, both in MCF-7 and MDA-MB-231 cells. The effect of the treatment of UA plus EPI in combination on these proteins was reversed by 3-MA. Based on these findings, we inferred that UA could enhance the EPI-induced autophagy and the changed level of Beclin-1 further indicated that UA was involved in the regulation of the class III PI3K(VPS34)/Beclin-1 pathway by EPI.

The class I PI3K/AKT/mTOR pathway is the major negative signaling pathway against autophagy [34]. PI3K/Akt is a classic upstream signal of mTOR and regulates mTOR activation [35]. Deng et al. reported that deactivating the PI3K/Akt/mTOR pathway suppresses renal carcinoma cancer cells by increasing autophagy [36]. Meng et al. found UA could inhibit cell proliferation of human prostate cancer cells through modulation of the PI3K/Akt/mTOR pathway [37]. Similar results were also found in our study: the treatment of UA plus EPI in combination could also markedly decrease the expression levels of PI3K and AKT compared with the UA-treated alone or the EPI-treated alone group both in MCF-7 and MDA-MB-231 cells. The decrease of these proteins induced by the treatment of UA plus EPI in combination was reversed by 3-MA both in MCF-7 and MDA-MB-231 cells. These results demonstrated that UA enhanced the regulation of activation of the PI3K/AKT/mTOR pathway by EPI.

Autophagy plays a critical role in cellular survival through reduced growth and increased catabolic lysis of unnecessary proteins and organelles when cancer cells are subjected to stressful conditions [16]. However, persistent or excessive autophagy also was reported to promote cell death following treatment with specific chemotherapeutic agents [38]. Despite the paradoxical roles of autophagy in tumor suppression and tumor promotion, manipulation of autophagic processes remains an attractive prospect as a new method of cancer therapy. In this study, we found that UA enhanced autophagy-induced by EPI and promoted autophagic cell death in MCF-7 and MDA-MB-231 cells. These findings provide evidence that the co-treatment of UA and EPI may be a new therapeutic strategy for breast cancer.

## 4. Materials and Methods

### 4.1. Reagents and Instruments

Dulbecco’s modified Eagle’s medium (DMEM) and fetal bovine serum (FBS) were purchased from GIBCO (Grand Island, NY, USA). Ursolic acid (UA) was purchased from Nanjing Zelang Plant Extract Co., Ltd. (Nanjing, China). Epirubicin was purchased from Rhawn (Shanghai, China). Autophagy inhibitor, 3-Methyladenine (3-MA), was purchased from Selleck Chemicals (Houston, TX, USA). Protease inhibitor, ECL luminescence reagent, Cell lysis buffer, Cell Counting Kit-8, Caspase3 ELISA Kit, Bradford Protein Assay Kit and autophagy agonist, rapamycin (RAPA), were obtained from Solarbio (Beijing, China). Class I PI3K, AKT, Beclin-1, LC3-II/LC3-, Atg5, Atg7 and β-Actin antibodies were purchased from CST (USA). RIPA Lysis Buffer (RIPA) was purchased from Biosharp (Guangzhou, China). Monodansylcadaverine (MDC) and Transwell chambers were purchased from Sigma (Shanghai, China). BCA Protein Assay Kit (BCA) was purchased from Bioteke (Beijing, China).

### 4.2. Cell Culture

The human breast cancer cells, MCF-7 and MDA-MB-231, were a kind gift from Professor Huanran Tan from Peking University Health Science Center. Cells were grown in DMEM supplemented with 10% FBS in a humidified incubator and flushed continuously at 37 °C with 5% CO_2_. When the cells reached 80% confluence, they were seeded onto 24-well or 6-well plates for further experiments. UA and EPI were dissolved in DMSO, respectively, and diluted into different concentrations with the culture medium. The final DMSO concentration in all assays did not exceed 0.1% (*v/v*).

### 4.3. Cell Viability Assessment

Cell viability assessment was performed by using Cell Counting Kit with the published methods [39], with some modifications. Cells were seeded in a 96-well plate at a density of 8 × 10^4^ cells/well. After 24 h, the medium was removed and a fresh medium along with various concentrations of EPI and/or UA (EPI: 0, 20, 40, 60, 80 μM; UA: 0, 1, 5, 10, 20 μM; 5, 10 μM UA plus 0.5, 1, 5 μM EPI respectively in combination in MCF-7 cells; 10, 20 μM UA plus 1, 5 μM EPI respectively in combination in MDA-MB-231 cells) was added to the cultures. 

After 24 h incubation, 10 μL CCK-8 was added to each well of the plate. Next, the plate was incubated for 2 h in the incubator at 37 °C with 5% CO_2_. Then, the absorbance was determined at 450 nm using an ELISA reader (TECAN Infinite 200 Pro, Tecan, Männedorf, Switzerland).

### 4.4. Caspase 3 Activity Assay

Cells were seeded in 6-well plates at a density of 1 × 10^6^ cells/well and incubated overnight. Then, the medium was replaced with fresh medium and the cells were treated with EPI and/or UA (10 μM UA and/or 1 μM EPI in MCF-7 cells; 20 μM UA and/or 1 μM EPI in MDA-MB-231 cells) for 24 h. Then cell lysis buffer was used to extract all protein from the cells. After that, lysates were cleared by centrifugation, followed by measurement of protein concentration by the Bradford method. Caspase-3 activity was evaluated using a commercially available Caspase3 ELISA Kit according to the manufacturer’s instructions. The absorbance values were determined at 405 nm using the ELISA plate reader. The activity levels were expressed compared with the control group.

### 4.5. Transwell Assay

The effect of EPI and/or UA on migration capabilities of MCF-7 and MDA-MB-231 cells was detected by Transwell assay. A cell suspension was re-suspended in serum-free DMEM medium, and 300 μL cell suspension was seeded in the upper part of the chamber at a density of 8 × 10^4^ cells/mL. Then, 500 μL DMEM medium containing 10% FBS was added to the lower parts of the chamber. After 24 h of EPI and/or UA (10 μM UA and/or 1 μM EPI in MCF-7 cells; 20 μM UA and/or 1 μM EPI in MDA-MB-231 cells) treatment, the cells were fixed with 4% paraformaldehyde for 1 h and then dyed with a 0.2% crystal violet solution for 15 min. Finally, the stained cells were washed and observed at ×200 under an inverted microscope.

### 4.6. The Effects of the Combination of EPI with UA on Autophagy in MCF-7 and MDA-MB-231 Cells

The cells were seeded in 96 well plates at a density of 8 × 10^4^ cells/well. After 24 h, the medium was removed and fresh medium added. The cells were treated with 3-MA (1 mM) or RAPA (10 ng/mL). The cells without treatment served as control. After 4 h incubation, the cells were treated with UA plus EPI in combination (10 μM UA plus 1 μM EPI in combination in MCF-7 cells; 20 μM UA plus 1 μM EPI in combination in MDA-MB-231 cells). After 22 h incubation, 10 μL CCK-8 was added to each well of the plate. Next, the plate was incubated for 2 h in the incubator at 37 °C with 5% CO_2_. Then, the absorbance was determined at 450 nm using an ELISA reader (TECAN Infinite 200 Pro, Männedorf, Switzerland).

### 4.7. Quantification of Monodansylcadaverine Cell Labeling

Monodansylcadaverine (MDC) is a specific autophagolysosome marker for analyzing the autophagic process. Cells were seeded in a 24-well plate at a density of 5 × 10^4^ cells/well for 24 h. Then, the medium was replaced with fresh medium and the cells were treated with UA plus EPI in combination (10 μM UA plus 1 μM EPI in combination in MCF-7 cells; 20 μM UA plus 1 μM EPI in combination in MDA-MB-231 cells) with/without 3-MA (1 mM) added 4 h in advance for 24 h. Then, autophagic vacuoles were labeled with MDC by incubating cells with 10 μM MDC at 37 °C for 20 min. After incubation, cells were washed 3 times with phosphate-buffered saline, and fluorescence changes at an excitation wavelength of 488 nm and an emission filter at 570 nm were observed by fluorescence microscopy (Olympus, Tokyo, Japan). 

### 4.8. Western Blot Analysis

Cells were seeded in a 6-well plate at a density of 8 × 10^4^ cells/well for 24 h. Then, the medium was replaced with a fresh medium and the cells were treated with various concentrations of EPI and/or UA (10 μM UA and/or 1 μM EPI in MCF-7 cells; 20 μM UA and/or 1 μM EPI in MDA-MB-231 cells with/without 3-MA (1 mM) or RAPA (10 ng/mL) added 4 h in advance) for 24 h. Total cell lysates were harvested by RIPA buffer containing complete protease inhibitors, and the total protein concentrations of the supernatants were assessed with a BCA kit. The equal amounts of the extracted proteins were loaded and subjected to SDS-polyacrylamide gels. After electrophoresis, the proteins were electrotransferred to polyvinylidene fluoride (PVDF) membranes. The membranes were blocked with 5% BSA and then incubated with diluted primary antibodies overnight at 4 °C. Then, the membranes were washed with 0.1% Tween-20 in Tris-buffered saline (TBS) and incubated with secondary antibodies for 2 h at room temperature with gentle shaking. Protein–antibody complexes were detected with the ECL substrate by the Chemdoc imaging system. Finally, the amounts of target proteins were quantified by ImageJ (NIH, Bethesda, MD, USA).

### 4.9. Statistical Analysis

Statistical significance was determined using GraphPad Prism 8.0 software (San Diego, CA, USA). All data were presented as the means ± SEM. Statistical analysis among multiple groups was carried out using one-way or two-way ANOVA followed by a Benjamini–Hochberg post-test. A *p*-value < 0.05 was considered statistically significant.

## 5. Conclusions

Overall, our research indicated that UA could be synergistic with EPI in the treatment of human breast cancer cells MCF-7 and MDA-MB-231. We further found that UA can significantly improve the drug sensitivity of human breast cancer MCF-7 and MDA-MB-231 cells to EPI. We also explored the potential mechanism of the enhanced effect of UA on the sensitivity of MCF-7 and MDA-MB-231 cells to EPI. Our results illustrated UA had effects on the class III PI3K(VPS34)/Beclin-1 pathway and PI3K/AKT/mTOR pathway to enhance the EPI-induced autophagy in MCF-7 and MDA-MB-231 cells. However, further in vivo studies are necessary to verify the enhanced effect of UA on the sensitivity of MCF-7 and MDA-MB-231 cells to EPI. Based on these findings, the combination strategy of UA and EPI may offer an alternative treatment for breast cancer.

## Figures and Tables

**Figure 1 molecules-27-03399-f001:**
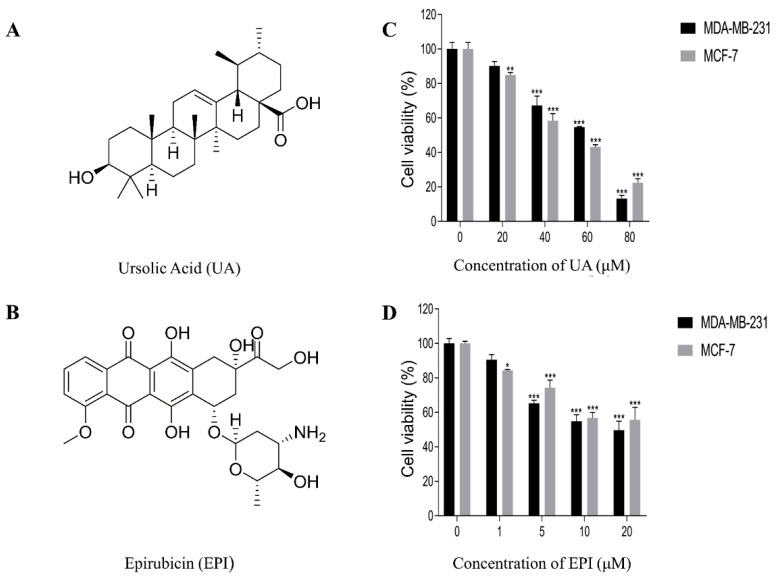
Chemical structures of Ursolic Acid (**A**) and Epirubicin (**B**). The effects of UA or EPI on cell viability were determined by CCK-8 assay. (**C**) MCF-7 and MDA-MB-231 cells were treated with the indicated concentrations of UA for 24 h. (**D**) MCF-7 and MDA-MB-231 cells were treated with the indicated concentrations of EPI for 24 h. The means and SEM were shown. *n* = 3. * *p* < 0.05, ** *p* < 0.01 and *** *p* < 0.001 versus cells without drug treatment.

**Figure 2 molecules-27-03399-f002:**
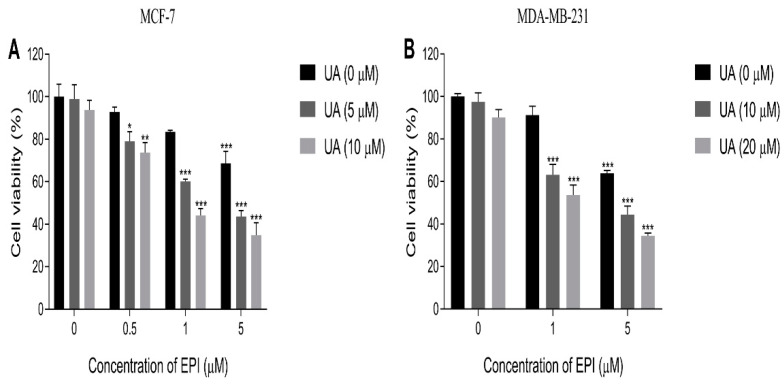
The effects of UA or EPI on cell viability were determined by CCK-8 assay. (**A**) MCF-7 and MDA-MB-231 cells were treated with the indicated concentrations of UA for 24 h. (**B**) MCF-7 and MDA-MB-231 cells were treated with the indicated concentrations of EPI for 24 h. The means and SEM were shown. *n* = 3. * *p* < 0.05, ** *p* < 0.01 and *** *p* < 0.001 versus cells without drug treatment in the absence of any treatments.

**Figure 3 molecules-27-03399-f003:**
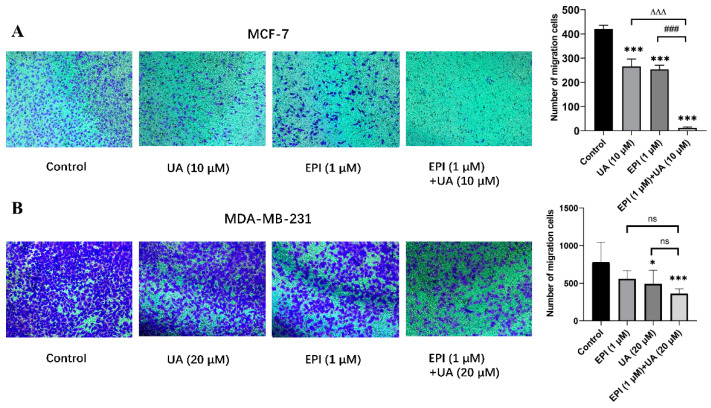
The effects of UA and/or EPI on the migration of MCF-7 and MDA-MB-231 cells. (**A**) MCF-7 cells were incubated with 10 μM of UA or 1 μM of EPI, or the two in combination for 24 h. (**B**) MDA-MB-231 cells were incubated with 20 μM of UA or 1 μM of EPI, or the two in combination for 24 h. The means and SEM were shown. *n* = 3. * *p* < 0.05 and *** *p* < 0.001 versus control group in the absence of any treatments; ### *p* < 0.001 versus EPI-treated alone group; ∆∆∆ *p* < 0.001 versus UA-treated alone group. The value bars with ns are not significant.

**Figure 4 molecules-27-03399-f004:**
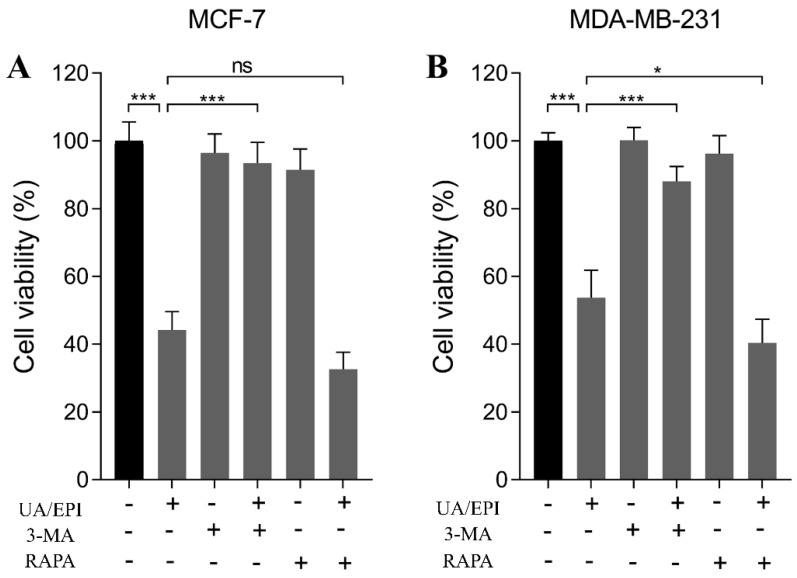
The effects of UA/EPI on autophagy of MCF-7 and MDA-MB-231 cells. MCF-7 (**A**) and MDA-MB-231 (**B**) cells were pre-treated with or without 3-MA (1 mM) or RAPA (10 ng/mL) for 4 h, followed by UA/EPI incubation for another 24 h. * *p* < 0.05 and *** *p* < 0.001. The value bars with ns are not significant.

**Figure 5 molecules-27-03399-f005:**
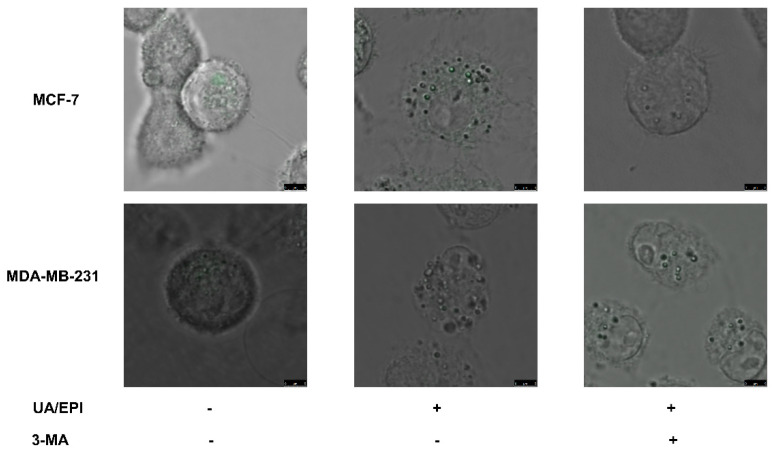
Detection of autophagy by MDC staining in breast cancer cells. MCF-7 and MDA-MB-231 cells were pre-treated with or without 3-MA (1 mM) for 4 h, followed by UA/EPI incubation for another 24 h, and then incubated with 0.05 mM monodansylcadaverine (MDC) for 1 h. The morphological changes were then examined by fluorescence microscopy. Scale bar = 5 μm. Autophagic vacuoles distributed within the cytoplasm or perinuclear regions stained by MDC appear as distinct dot-like structures.

**Figure 6 molecules-27-03399-f006:**
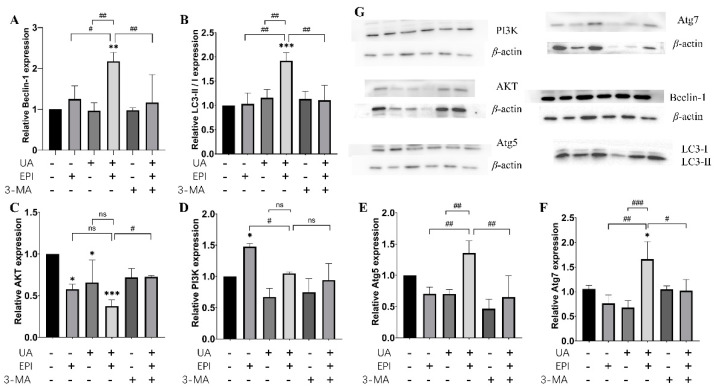
Changes in the expression levels of autophagy-related protein in MCF-7 cells. Western blot analysis was used to measure autophagy-related protein expression levels in MCF-7 cells. (**A**) Quantification of Beclin-1 protein expression levels in MCF-7 cells. (**B**) Quantification of LC3-II/I protein expression levels in MCF-7 cells. (**C**) Quantification of AKT protein expression levels in MCF-7 cells. (**D**) Quantification of PI3K protein expression levels in MCF-7 cells. (**E**) Quantification of Atg5 protein expression levels in MCF-7 cells. (**F**) Quantification of Atg7 protein expression levels in MCF-7 cells. (**G**) Representative pictures of Beclin-1, LC3-II/I, AKT, PI3K, Atg5, Atg7 expression by Western blot. The means and SEM were shown. *n* = 3. * *p* < 0.05, ** *p* < 0.01 and *** *p* < 0.001 versus control group in the absence of any treatments; # *p* < 0.05, ## *p* < 0.01 and ### *p* < 0.001. The value bars with ns are not significant.

**Figure 7 molecules-27-03399-f007:**
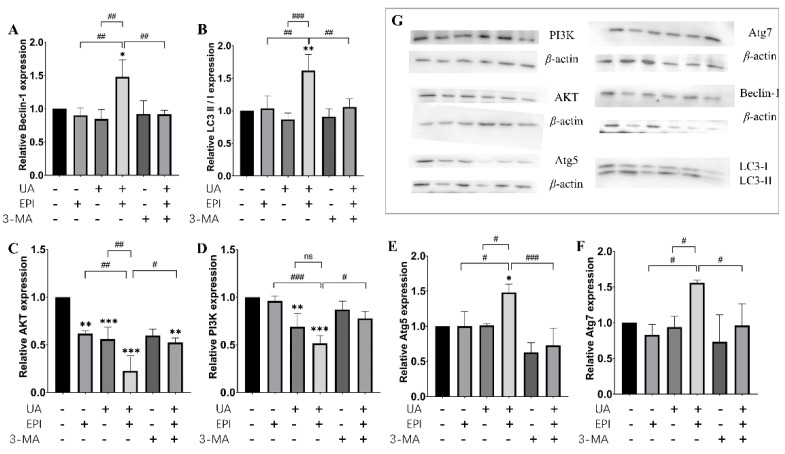
Changes in the expression levels of autophagy-related protein in MDA-MB-231 cells. Western blot analysis was used to measure autophagy-related protein expression levels in MDA-MB-231 cells. (**A**) Quantification of Beclin-1 protein expression levels in MDA-MB-231 cells. (**B**) Quantification of LC3-II/I protein expression levels in MDA-MB-231 cells. (**C**) Quantification of AKT protein expression levels in MDA-MB-231 cell. (**D**) Quantification of PI3K protein expression levels in MDA-MB-231 cells. (**E**) Quantification of Atg5 protein expression levels in MDA-MB-231 cells. (**F**) Quantification of Atg7 protein expression levels in MDA-MB-231 cells. (**G**) Representative pictures of Beclin-1, LC3-II/I, AKT, PI3K, Atg5, Atg7 expression by Western blot. The means and SEM were shown. *n* = 3. * *p* < 0.05, ** *p* < 0.01 and *** *p* < 0.001 versus control group in the absence of any treatments; # *p* < 0.05, ## *p* < 0.01 and ### *p* < 0.001. The value bars with ns are not significant.

**Table 1 molecules-27-03399-t001:** The percentage of MCF-7 viability was affected by ursolic acid, epirubicin, and ursolic acid combined with epirubicin^1^. Data represent means ±SEM (*n* = 3).

Agents	Concentration (µg/mL)/Cell Viability (%)		
	UA (0 μM)	UA (5 μM)	UA (10 μM)
EPI (0 μM)	100 ± 5.84	98.91 ± 6.79	93.78 ± 4.53
EPI (0.5 μM)	92.83 ± 2.37	79.02 ± 4.54 *_#_	73.76 ± 4.74 **_#__∆_
EPI (1 μM)	83.40 ± 0.83	60.10 ± 1.13 ***_###__∆∆_	44.19 ± 3.14 ***_###__∆∆∆_
EPI (5 μM)	68.67 ± 5.69 ***	43.63 ± 2.82 ***_###__∆∆_	34.85 ± 5.90 ***_###__∆∆∆_

^1^*p*-values corrected by BH (Benjamini Hochberg) methods were calculated based on a parametric Student’s *t*-test or a nonparametric Mann–Whitney test (dependent on the conformity to normal distribution). * *p* < 0.05, ** *p* < 0.01 and *** *p* < 0.001 versus control group in the absence of any treatments; _#_
*p* < 0.05 and _###_
*p* < 0.001 versus UA-treated alone group; _∆_
*p* < 0.05, _∆∆_
*p* < 0.01, _∆∆∆_
*p* < 0.001 versus EPI- treated alone group.

**Table 2 molecules-27-03399-t002:** The percentage of MDA-MB-231 viability was affected by ursolic acid, epirubicin, and ursolic acid combined with epirubicin^1^. Data represent means ±SEM (*n* = 3).

Agents	Concentration (µg/mL)/Cell Viability (%)		
	UA (0 μM)	UA (10 μM)	UA (20 μM)
EPI (0 μM)	100 ± 1.40	97.50 ± 4.23	90.12 ± 3.70
EPI (1 μM)	91.24 ± 4.26	63.12 ± 4.91 ***_###__∆∆∆_	53.68 ± 4.70 ***_###__∆∆∆_
EPI (5 μM)	63.80 ± 1.48 ***	44.49 ± 3.99 ***_###__∆∆_	34.49 ± 1.24 ***_###__∆∆∆_

^1^*p*-values corrected by BH (Benjamini Hochberg) methods were calculated based on a parametric Student’s *t*-test or a nonparametric Mann–Whitney test (dependent on the conformity to normal distribution). *** *p* < 0.001 versus control group in the absence of any treatments; _###_
*p* < 0.001 versus UA-treated alone group; _∆∆_
*p* < 0.01 and _∆∆∆_
*p* < 0.001 versus EPI-treated alone group.

## Data Availability

The datasets used and/or analyzed during the current study are available from the corresponding author on reasonable request.

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
