# Peer review of "Ursolic Acid Enhances the Sensitivity of MCF-7 and MDA-MB-231 Cells to Epirubicin by Modulating the Autophagy Pathway"

_molecules, 2022, doi:10.3390/molecules27113399_

Round 1

Reviewer 1 Report

The manuscript by Wang et al “Ursolic acid enhances the sensitivity of MCF-7 and MDA-MB- 231 cells to epirubicin by modulating the autophagy pathway” provides interesting data on the the synergistic effect of ursolic acid with epirubicin in the breast cancer treatment. There are several concerns that are summarized below before publication is possible.

  1. Authors did not mention Fig 1A and 1B anywhere in the manuscript.
  2. Remove % in the table and mention it in the legend as viability presented in %.
  3. Figure 1 legend mention as “cells without drug treatment” instead of control in the statistical part.
  4. Section 2.2 it will be good if authors do provide the data for 10 μM of UA on MDA-MB-231 and 5, 10 μM of UA on MCF-7 viability assays.
  5. Fix the typo errors in figure 2 legend, it doesn’t have any structures as mentioned.
  6. Fix the typo errors in figure 3 and name it as control and not as con. Also, provide the drug concentration in the figure itself. Compare the statistical difference between UA and EPI + UA group as well. Explain # in the legend.
  7. Authors should explain how they claim the combinatorial therapy in figure 4 with UA/EPI + 3-MA restores the viability. Since treatment with 3-MA has similar viability in the pre-treatment step.
  8. Figure 6 and 7 densitograms doesn’t correlates with the Westerns presented in the figure. I would suggest to re-quantify. Visually there is difference in the Beclin blot and looks like actin. While actin blot looks like Beclin authors should clarify this.
  9. Section 4.6 mention the conc of 3-MA and RAPA used.
  10. Include few more references in the discussion section.
  11. There are several typo errors throughout the manuscript.

Author Response

  1. Authors did not mention Fig 1A and 1B anywhere in the manuscript.

Respond: Thank you for your reminding. Figures 1A and 1B were added in the Introduction part to show clearly the structures of Ursolic acid and Epirubicin, as follows:

Lines 30-31, Page 1: Epirubicin (EPI) (Figure 1B) is an anthracycline drug commonly used in the clinical treatment of breast cancer.

Lines 50-51, Page 2: Ursolic acid (UA) (Figure 1A), a pentacyclic triterpene, is a common chemical constituent existing mainly in the Lamiaceae family.

  1. Remove % in the table and mention it in the legend as viability presented in %.

Respond: Thank you, the % in the table was removed and mentioned in the legend as viability presented in % in Table1 and Table 2.

  1. Figure 1 legend mention as “cells without drug treatment” instead of control in the statistical part.

Respond: Thank you for your nice comments on our article. The Figure 1 legend was rewritten, as follows:

Lines 83-87, Page 3: Figure 1 Chemical structures of Ursolic Acid (A) and Epirubicin (B). The effects of UA or EPI on cell viability were determined by CCK-8 assay. (C) MCF-7 and MDA-MB-231 cells were treated with the indicated concentrations of UA for 24 h. (D) MCF-7 and MDA-MB-231 cells were treated with the indicated concentrations of EPI for 24 h. The means and SEM were shown. n = 3. *P < 0.05, **P < 0.01 and ***P < 0.001 versus cells without drug treatment..

  1. Section 2.2 it will be good if authors do provide the data for 10 μM of UA on MDA-MB-231 and 5, 10 μM of UA on MCF-7 viability assays.

Respond: Thank the reviewer for the constructive comments. The data for 10 μM of UA on MDA-MB-231 and 5, 10 μM of UA on MCF-7 viability assays, as follows:

Lines 95-99, Page 3: When treated with the 5, 10 μM UA alone, the viability of MCF-7 cells was 98.91% and 93.78%, respectively, which indicated that UA had no significant effects on the viability of MCF-7 cells. And the viability of MCF-7 cells was decreased to 92.83%, 83.40%, 68.67%, respectively, when treated which EPI alone at concentrations of 0.5 μM, 1 μM and 5 μM.

Lines 104-108, Page 3: UA showed little inhibitory effect on the viability of MDA-MB-231 cells at the concentrations of 10 μM, 20 μM (the viability of MDA-MB-231 cells was 97.50%, 90.12%, respectively). And UA also showed a remarkable enhanced ability on the sensitivity of MDA-MB-231 cells to EPI when treated with 20 μM UA plus 1 μM EPI in combination

  1. Fix the typo errors in figure 2 legend, it doesn’t have any structures as mentioned.

Respond: Many thank you for your comments. The Figure 2 legend was rewritten, as follows:

Lines 129-133, Page 4: The effects of UA or EPI on cell viability were determined by CCK-8 assay. (A) MCF-7 and MDA-MB-231 cells were treated with the indicated concentrations of UA for 24 h. (B) MCF-7 and MDA-MB-231 cells were treated with the indicated concentrations of EPI for 24 h. The means and SEM were shown. n = 3. *P < 0.05, **P < 0.01 and ***P < 0.001 versus control group in the absence of any treatments.

  1. Fix the typo errors in figure 3 and name it as control and not as con. Also, provide the drug concentration in the figure itself. Compare the statistical difference between UA and EPI + UA group as well. Explain # in the legend.

Respond: We sincerely appreciate the valuable comments. The errors in figure 3 were corrected and provided the drug concentration in the figure. the statistical difference between UA and EPI + UA group was compared, and explained them in the legend, as follows:

Lines 144-149, Page 5: Figure 3 The effects of UA and/or EPI on the migration of MCF-7 and MDA-MB-231 cells. (A) MCF-7 cells were incubated with 10 μM of UA or 1 μM of EPI, or the two in combination for 24h. (B) MDA-MB-231 cells were incubated with 20 μM of UA or 1 μM of EPI, or the two in combination for 24 h. The means and SEM were shown. n = 3. *P < 0.05, **P < 0.01 and ***P < 0.001 versus control group in the absence of any treatments; # P < 0.05, ##P < 0.01 and ### P < 0.001 versus EPI-treated alone group; ∆ P<0.05,∆∆ P<0.01,∆∆∆ P<0.001 versus UA-treated alone group.

  1. Authors should explain how they claim the combinatorial therapy in figure 4 with UA/EPI + 3-MA restores the viability. Since treatment with 3-MA has similar viability in the pre-treatment step.

Respond: We sincerely appreciate the valuable comments and rewrote the sentence in the context to make our points clearer, as follows:

Lines 156-160, Page 5: As shown in Figure 4A, the viability of MCF-7cells was decreased significantly when treated with 10 μM UA plus 1 μM EPI in combination. Whereas, 3-MA (1mM), the autophagy inhibitor, which had no remarkable effect on the viability of MCF-7cells, could significantly reverse the inhibitory effect of treatment with 10 μM UA plus 1 μM EPI in combination in MCF-7 cells and the viability of cells was restored to 93.43%.

Lines 162-164, Page 5: A similar result could be found in MDA-MB-231 cells. 3-MA (1mM) restored MDA-MB-231 cells viability to 88.01%, attenuating the inhibitory effect of treatment with 20 μM UA plus 1 μM EPI in combination (Figure 4B).

  1. Figure 6 and 7 densitograms doesn’t correlates with the Westerns presented in the figure. I would suggest to re-quantify. Visually there is difference in the Beclin blot and looks like actin. While actin blot looks like Beclin authors should clarify this.

Respond: Thank the reviewer for the comments. In this, study, there were indeed differences in internal reference, β-actin blot, among groups, which may be induced by personal equation. Nevertheless, it would not affect our results. Because, the expression of the autophagy-related protein in western blot analysis was estimated by the ratio of autophagy-related protein/β-action, which would eliminate the effects of the personal equation. So, our results are credible to support our points that UA promotes EPI-sensitivity of MCF-7/MDA-MB-231 cells by modulating the autophagy pathway.

  1. Section 4.6 mention the conc of 3-MA and RAPA used.

Respond: Thank you for your reminding. The concentrations of 3-MA and RAPA were added in Section 4.6, as follows:

Lines 361-362, Page 11: The cells were treated with 3-MA (1 mM) or RAPA (10 ng/mL). The cells without treatment served as control.

  1. Include few more references in the discussion section.

Respond: Thank the reviewer for the comments. We added some more references in our manuscript to make our points more detailed and clear, and copied them as follows:

Lines 244-246, Page 9: Xavier et al found UA could modulate autophagy through the JNK pathway in apoptosis-resistant colorectal cancer cells and induce cell death [24].

Lines 270-273, Page 9: Leng, et al found UA promotes cancer cell death by inducing Atg5-dependent autophagy [30]. Additionally, UA also had been demonstrated to cause the upregulation of Beclin-1 in hypertrophic scar fibroblasts [31]. Similar results were also found in our study.

Lines 284-286, Page 9: And Meng et al found UA could inhibit cell proliferation of human prostate cancer cells through modulation of the PI3K/Akt/mTOR pathway [35].

Reference:

  1. Xavier, C.P.R.; Lima, C.F.; Pedro, D.F.N.; Wilson, J.M.; Kristiansen, K.Pereira-Wilson, C., Ursolic acid induces cell death and modulates autophagy through JNK pathway in apoptosis-resistant colorectal cancer cells. J Nutr Biochem 2013, 24, 706-712.
  2. Leng, S.; Hao, Y.; Du, D.; Xie, S.; Hong, L.; Gu, H.; Zhu, X.; Zhang, J.; Fan, D.Kung, H.-f., Ursolic acid promotes cancer cell death by inducing Atg5-dependent autophagy. Int J Cancer 2013, 133, 2781-2790.
  3. Cao, C.; Wang, W.; Lu, L.; Wang, L.; Chen, X.; Guo, R.; Li, S.Jiang, J., Inactivation of Beclin-1-dependent autophagy promotes ursolic acid-induced apoptosis in hypertrophic scar fibroblasts. Exp Dermatol 2018, 27, 58-63.
  4. Meng, Y.; Lin, Z.-M.; Ge, N.; Zhang, D.-L.; Huang, J.Kong, F., Ursolic Acid Induces Apoptosis of Prostate Cancer Cells via the PI3K/Akt/mTOR Pathway. Am J Chinese Med 2015, 43, 1471-1486.

  1. There are several typo errors throughout the manuscript.

Respond: We feel sorry for our carelessness. We had checked all of the manuscript and corrected all typo errors, as follows:

Line 236, Page 8: ‘a therapy’ was corrected to ‘therapy’.

Line 258, Page 9: ‘network’ was corrected to ‘networks’.

Line 301, Page 9: ‘evidences’ was corrected to ‘evidence’

Line 36, Page1 and Line 301, Page 9: ‘therapy’ was corrected to ‘therapeutic’.

Lines 350-351, Page 10: ‘cells suspension’ was corrected to ‘cell suspension’.

Reviewer 2 Report

This paper explores the effect of the natural product steroid ursolic acid (UA) on the inhibitory properties of the anthracycline epirubicin towards breast cancer cell lines. The authors note that UA “enhances cell sensitivity to epirubicin by modulating the autophagy pathway”. It is already well-known (multiple papers; e.g., Xavier et al,, J Nutr Biochem 2013 24:706; Leng, et al., Cancer Cell Biology. 2013, 133; Lewinska et al. Apoptosis, 2017, 22, 800; none of which are reported here) that UA promotes autophagy and enhances the effects of anthracyclines in cancer cell lines. In this aspect the paper is not novel. Thus, while the authors finally note that “Our study may provide a new therapeutic strategy for combination therapy”, but I suggest this had already been done.

Nevertheless, the present manuscript is a detailed and well-executed study that adds some new points; that UA enhances the inhibitory effect of epirubicin on cell migration and on the regulation of activation of the PI3K(VPS34)/Beclin-1 and PI3K/AKT/mTOR pathways. But it needs to include some reference to the earlier work that delineated these pathways.

Author Response

Respond: Thank you for the constructive comments. We rewrote these sentences in the context and added some more references in our manuscript to make our points more detailed and clear, and copied them as follows:

To the best of our knowledge, we have revised all the points you gave. I think all of the revising points would fit your suggestions.

Lines 54-56, Page 2: Lewinska et al. found UA could mediate changes in the glycolytic pathway to promote cytotoxic autophagy and apoptosis in phenotypically different breast cancer cells [15].

Lines 66-67, Page 2: These findings provide evidence that UA might be an effective adjunct to EPI in the treatment of breast cancer.

Lines 244-246, Page 9: Xavier et al found UA could modulate autophagy through the JNK pathway in apoptosis-resistant colorectal cancer cells and induce cell death [24].

Lines 270-273, Page 9: Leng, et al found UA promotes cancer cell death by inducing Atg5-dependent autophagy [30]. Additionally, UA also had been demonstrated to cause the upregulation of Beclin-1 in hypertrophic scar fibroblasts [31]. Similar results were also found in our study.

Lines 284-286, Page 9: And Meng et al found UA could inhibit cell proliferation of human prostate cancer cellsthrough modulation of the PI3K/Akt/mTOR pathway [35].

Reference:

15. Lewinska, A.; Adamczyk-Grochala, J.; Kwasniewicz, E.; Deregowska, A.Wnuk, M., Ursolic acid-mediated changes in glycolytic pathway promote cytotoxic autophagy and apoptosis in phenotypically different breast cancer cells. Apoptosis 2017, 22, 800-815

24. Xavier, C.P.R.; Lima, C.F.; Pedro, D.F.N.; Wilson, J.M.; Kristiansen, K.Pereira-Wilson, C., Ursolic acid induces cell death and modulates autophagy through JNK pathway in apoptosis-resistant colorectal cancer cells. J Nutr Biochem 2013, 24, 706-712.

30. Leng, S.; Hao, Y.; Du, D.; Xie, S.; Hong, L.; Gu, H.; Zhu, X.; Zhang, J.; Fan, D.Kung, H.-f., Ursolic acid promotes cancer cell death by inducing Atg5-dependent autophagy. Int J Cancer 2013, 133, 2781-2790.

31. Cao, C.; Wang, W.; Lu, L.; Wang, L.; Chen, X.; Guo, R.; Li, S.Jiang, J., Inactivation of Beclin-1-dependent autophagy promotes ursolic acid-induced apoptosis in hypertrophic scar fibroblasts. Exp Dermatol 2018, 27, 58-63.

35. Meng, Y.; Lin, Z.-M.; Ge, N.; Zhang, D.-L.; Huang, J.Kong, F., Ursolic Acid Induces Apoptosis of Prostate Cancer Cells via the PI3K/Akt/mTOR Pathway. Am J Chinese Med 2015, 43, 1471-1486.

Round 2

Reviewer 1 Report

Authors have addressed my comments but not all. 

I'm not satisfied with the answers for my initial comment regarding Fig 4 combinatorial therapy and Western blots data.

Authors should clarify this. 

Still error in the Fig 3 is not fixed see MCF-7 presented as MFC-7

Author Response

Response to Reviewer 1:

Authors have addressed my comments but not all. 

I'm not satisfied with the answers for my initial comment regarding Fig 4 combinatorial therapy and Western blots data.

Authors should clarify this. 

Still error in the Fig 3 is not fixed see MCF-7 presented as MFC-7

  1. The initial comment regarding Fig 4 combinatorial therapy: Authors should explain how they claim the combinatorial therapy in figure 4 with UA/EPI + 3-MA restores the viability. Since treatment with 3-MA has similar viability in the pre-treatment step.

Respond: Many thank you for your comments. As shown in the figure 4, 3-MA (1mM), the autophagy inhibitor, which showed no remarkable effect on the viability of MCF-7 and MCF-7 MDA-MB-231 cells. And UA plus EPI in combination could markedly decrease the viabilities of MCF-7 and MDA-MB-231 cells compared with the control group in the absence of any treatments. However, the inhibitory effects of the treatment with 10 μM UA plus 1 μM EPI could be reversed by 3-MA, the viabilities of MCF-7 and MDA-MB-231 cells was restored to 93.43% and 88.01%, respectively. These results may be the reason why the viabilities of MCF-7 and MDA-MB-231 cells in UA/EPI + 3-MA group were similar to those in 3- MA alone group.

This assay aimed to explore whether the effects of the treatment of UA plus EPI in combination in MCF-7 and MDA-MB-231 cells were related to the autophagy pathway. So we added 3-MA (Autophagy inhibitor) or RAPA (autophagy agonist) to cells for 4 hours before the treatment of UA plus EPI in combination to inhibit or activate cell autophagy. And then we would detect whether the inhibitory effects on the treatment of UA plus EPI in combination in MCF-7 and MDA-MB-231 cells were changed because of 3-MA or RAPA being added. If so, we would conclude that the effects of the treatment of UA plus EPI in combination in MCF-7 and MDA-MB-231 cells were related to the autophagy pathway.

And the results were as exhibited in figure 4, UA plus EPI in combination could markedly decrease the viabilities of MCF-7 and MDA-MB-231 cells compared with the control group in the absence of any treatments. Additionally, 3-MA (1mM), the autophagy inhibitor, which showed no remarkable effect on the viability of MCF-7 and MCF-7 MDA-MB-231 cells. Whereas, we found the viabilities MCF-7 and MDA-MB-231 cells were increased significantly after the treatment of 3-MA+ UA plus EPI in combination compared with the UA plus EPI in combination group, which indicated 3-MA (1mM) could significantly reverse the inhibitory effect of treatment with 10 μM UA plus 1 μM EPI in combination in MCF-7 cells and MDA-MB-231 cells.

These results are consistent with those of previous studies. Pan et al found the inhibitory effect of bufalin on colorectal cancer cells was reversed by treatment with 3-MA [17]. Lee et al. reported that 3-MA reversed the cytotoxic autophagic cell death induced by the treatment with combination of pristimerin and paclitaxel in MDA-MB-231 cells [18]. Based on these findings, we inferred that the inhibitory effects of UA plus EPI in combination in MCF-7 and MDA-MB-231 cells were related to the autophagy pathway.

And we rewrote these sentences in the context and added some related references to make our study easier to understand, as follows:

Lines 156-174, Pages 5-6: As shown in Figure 4A, the viability of MCF-7cells was decreased significantly when treated with 10 μM UA plus 1 μM EPI in combination compared with the control group in the absence of any treatments. And 3-MA (1mM), the autophagy inhibitor, which showed no remarkable effect on the viability of MCF-7cells. Whereas, we found the viabilities of MCF-7 cells were increased significantly after the treatment of 3-MA+ UA plus EPI in combination compared with the UA plus EPI in combination group, which indicated 3-MA (1mM) could significantly reverse the inhibitory effect of treatment with 10 μM UA plus 1 μM EPI in combination in MCF-7 cells and the viability of cells was restored to 93.43%. Additionally, RAPA influenced the inhibitory effect of treatment with 10 μM UA plus 1 μM EPI in combination little in MCF-7 cells. A similar result could be found in MDA-MB-231 cells. 3-MA (1mM) restored MDA-MB-231 cells viability to 88.01%, attenuating the inhibitory effect of treatment with 20 μM UA plus 1 μM EPI in combination (Figure 4B). And RAPA also slightly enhanced the inhibitory effect of UA and EPI combination treatment. These results are consistent with those of previous studies. Pan et al found the inhibitory effect of bufalin on colorectal cancer cells was reversed by treatment with 3-MA [17]. Lee et al. reported that 3-MA reversed the cytotoxic autophagic cell death induced by the treatment with combination of pristimerin and paclitaxel in MDA-MB-231 cells [18]. Based on these findings, we inferred that the inhibitory effects of UA plus EPI in combination in MCF-7 and MDA-MB-231 cells were related to the autophagy pathway.

We appreciate for Reviewer’s warm work earnestly and hope that the correction will meet with approval.

  1. Pan, Z.; Xie, Y.; Bai, J.; Lin, Q.; Cui, X.Zhang, N., Bufalin suppresses colorectal cancer cell growth through promoting autophagy in vivo and in vitro. Rsc. Adv. 2018, 8, 38910-38918.
  2. Lee, Y.; Na, J.; Lee, M.S.; Cha, E.Y.; Sul, J.Y.; Park, J. B.Lee, J.S., Combination of pristimerin and paclitaxel additively induces autophagy in human breast cancer cells via ERK1/2 regulation. Mol. Med. Rep. 2018, 18, 4281-4288.

  1. The initial comment regarding Western blots data: Figure 6 and 7 densitograms doesn’t correlates with the Westerns presented in the figure. I would suggest to re-quantify. Visually there is difference in the Beclin blot and looks like actin. While actin blot looks like Beclin authors should clarify this.

Respond: Thank the reviewer for the comments. We have re-quantified all the Western blots, and redrawn figures 6 and figure 7 to show the results. And the quantitative results are consistent with the points of view in this manuscript.

  1. Still error in the Fig 3 is not fixed see MCF-7 presented as MFC-7

Respond: Thank the reviewer for the comments. The error in Figure 3 was corrected.
